# Fucoxanthin Suppresses Osteoclastogenesis via Modulation of MAP Kinase and Nrf2 Signaling

**DOI:** 10.3390/md19030132

**Published:** 2021-02-27

**Authors:** You-Jung Ha, Yong Seok Choi, Ye Rim Oh, Eun Ha Kang, Gilson Khang, Yong-Beom Park, Yun Jong Lee

**Affiliations:** 1Division of Rheumatology, Department of Internal Medicine, Seoul National University Bundang Hospital, Seongnam 13620, Korea; hayouya@snubh.org (Y.-J.H.); kangeh@snubh.org (E.H.K.); 2Medical Science Research Institute, Seoul National University Bundang Hospital, Seongnam 13620, Korea; choicell1@hanmail.net (Y.S.C.); ptojt93@naver.com (Y.R.O.); 3Department of Bionanotechnology and Bio-Convergence Engineering, Department of PolymerNano Science and Technology and Polymer Materials Fusion Research Center, Chonbuk National University, Jeonju-si 54896, Korea; gskhang@jbnu.ac.kr; 4Division of Rheumatology, Department of Internal Medicine, Yonsei University College of Medicine, Seoul 03722, Korea; yongbpark@yuhs.ac; 5Department of Internal Medicine, Seoul National University College of Medicine, Seoul 03080, Korea

**Keywords:** brown seaweed, fucoxanthin, osteoclastogenesis, MAP kinase, Nrf2

## Abstract

Fucoxanthin (FX), a natural carotenoid present in edible brown seaweed, is known for its therapeutic potential in various diseases, including bone disease. However, its underlying regulatory mechanisms in osteoclastogenesis remain unclear. In this study, we investigated the effect of FX on osteoclast differentiation and its regulatory signaling pathway. In vitro studies were performed using osteoclast-like RAW264.7 cells stimulated with the soluble receptor activator of nuclear factor-κB ligand or tumor necrosis factor-alpha/interleukin-6. FX treatment significantly inhibited osteoclast differentiation and bone resorption ability, and downregulated the expression of osteoclast-specific markers such as nuclear factor of activated T cells 1, dendritic cell-specific seven transmembrane protein, and matrix metallopeptidase 9. Intracellular signaling pathway analysis revealed that FX specifically decreased the activation of the extracellular signal-regulated kinase and p38 kinase, and increased the nuclear translocation of phosphonuclear factor erythroid 2-related factor 2 (Nrf2). Our results suggest that FX regulates the expression of mitogen-activated protein kinases and Nrf2. Therefore, FX is a potential therapeutic agent for osteoclast-related skeletal disorders including osteoporosis and rheumatoid arthritis.

## 1. Introduction

Bones are a dynamic tissue that undergoes constant renewal and repair through bone remodeling. This process is characterized by the spatiotemporal coupling of osteoclast-induced bone resorption and osteoblast-induced bone formation. An imbalance between bone resorption and bone formation, especially excessive osteoclastic activity, is involved in the pathogenesis of osteoporosis, rheumatoid arthritis, multiple myeloma, and metastatic cancers [1,2]. Osteoclasts are multinucleated cells that are differentiated from hematopoietic precursor cells of monocyte or macrophage lineage by canonical stimulation with macrophage colony-stimulating factor (M-CSF) and receptor activator of nuclear factor-κB (NF-κB) ligand (RANKL) [3,4]. RANKL binds to its receptor on osteoclast precursors to activate the mitogen-activated protein kinase (MAPK) signaling pathway, and downstream transcription factors and osteoclast differentiation markers, including AP-1, NF-κB, and nuclear factor of activated T cells 1 (NFATc1) [2,5,6]. Some cytokines, such as tumor necrosis factor (TNF)-α and interleukin (IL)-6, can serve as noncanonical osteoclastogenic effectors in a RANKL-independent mechanism [7,8]. 

Fucoxanthin (FX) is an oxygenated carotenoid present in edible brown sea algae such as kombu (*Laminaria japonica*), wakame (*Undaria pinnatifida*), and arame (*Eisenia bicyclis*) [9]. Previous studies demonstrated that FX possesses antiobesity, antidiabetic, anti-inflammatory, anticancer, and hepatoprotective activities, in addition to its cerebrovascular protective effects [10,11,12,13,14,15,16,17,18]. Fruit carotenoids such as lycopene and cryptoxanthin were reported to inhibit osteoclastogenesis [19,20]. However, the effect and underlying mechanism of FX on osteoclastogenesis remain poorly understood. 

There is interest in the effect of FX on osteoclast differentiation. Although FX exhibited a limited antiosteoresorptive effect in a ligature-induced periodontitis mouse model, FX administration significantly reduced the number of RANKL-positive osteoclasts [21]. Das et al. [22] showed that the treatment of osteoclast-like RAW264.7 cells with 2.5 μM FX inhibits RANKL-induced osteoclast differentiation through an induction of apoptosis. Despite this, an extremely high dose of FX did not produce significant side effects in animal models. Moreover, several studies demonstrated that FX inhibits apoptosis or promotes the survival of various nonmalignant cells at concentrations of up to 50 μM [23,24,25,26]. Taira et al. [27] reported FX-induced cytotoxicity in RAW264.7 cells at 20 μM. Therefore, the underlying mechanisms of FX effects on the canonical and noncanonical osteoclastogenic signaling pathways could not depend on cellular apoptosis and are yet to be elucidated. In the present study, we investigated the effects of FX on RANKL-dependent and -independent osteoclast differentiation, and identified its molecular regulatory mechanisms.

## 2. Results

### 2.1. FX Effect on RAW264.7 Cell Viability

The cytotoxic effect of FX on RAW264.7 cells was determined using the 3-(4,5-dimethylthiazol-2-yl) -2,5-diphenyltetrazolium bromide (MTT) assay (Figure 1A). FX did not affect cell viability at a concentration of ≤5 μM. However, the number of viable cells after treatment with 10 μM of FX was significantly lower than the number of untreated cells was. Additionally, in contrast to the results of Das et al. [22], no cleavage of procaspase-3 and poly ADP ribose polymerase (PARP; Figure 1B) was detected in cells treated with ≤10 μM FX. Therefore, FX was used at a concentration of ≤5 μM in all subsequent experiments.

### 2.2. FX Inhibits Osteoclastogenesis

Tartrate-resistant acid phosphatase (TRAP) is highly expressed in differentiated osteoclasts, and is therefore used as a primary marker of osteoclastogenesis [28]. The treatment of soluble RANKL (sRANKL, 50 ng/mL) or costimulation with TNF-α (50 ng/mL) and IL-6 (50 ng/mL) in RAW264.7 cells increased the number of TRAP-positive and multinucleated cells compared with that of untreated control cells (Figure 2A). Under these conditions, FX treatment decreased the number of TRAP-positive multinucleated cells in a dose-dependent manner (Figure 2B). In experiments using human CD14+ monocytes, 0 to 5 μM of FX dose-dependently suppressed RANKL- and TNF-α/IL-6-induced osteoclast differentiation from osteoclast precursors (Figure 2C). These results suggest that FX inhibits the differentiation of RAW264.7 cells and human CD14+ monocytes to osteoclast-like cells.

The resorption pit assay was performed to examine osteoclast activity. As shown in Figure 3, the resorption pit area was significantly decreased upon treatment with FX in a dose-dependent manner. These results collectively suggest that FX exerts an inhibitory effect on both sRANKL-dependent and -independent bone-resorbing osteoclast activity.

### 2.3. FX Downregulates Osteoclast-Specific Markers and Transcriptional Factors in RAW264.7 Cells

NFATc1 is a master transcription factor for osteoclastogenesis [29]. The expression of NFATc1 in sRANKL- or TNF/IL-6-stimulated RAW264.7 cells decreased upon treatment with FX in a dose-dependent manner (Figure 4A). Dendritic-cell-specific transmembrane protein (DC-STAMP), another essential mediator for osteoclastogenesis, is upregulated upon osteoclastogenic stimulation. Increased DC-STAMP expression, in turn, upregulates the expression of osteoclast-specific markers such as *TRAP* [30]. As shown in Figure 4B, *DC-STAMP* mRNA expression was significantly decreased upon treatment with FX in a dose-dependent manner.

Matrix metallopeptidase (MMP)-9 is a well-established proteolytic effector of osteoclast-mediated bone resorption. To examine whether FX affects MMP-9 production in differentiated osteoclast-like cells, MMP-9 levels in culture supernatant were measured by ELISA. As shown in Figure 5, cells treated with 5 μM FX showed a significant decrease in MMP-9 levels. Moreover, MMP-9 production decreased following FX treatment in a dose-dependent manner.

To better understand the mechanism of the FX-induced inhibition of osteoclast differentiation, we performed immunoblot analysis of RAW264.7 cells to measure the expression of molecules known to be critically involved in the osteoclast signaling pathway, including MAPKs (extracellular signal-regulated kinase (ERK), c-Jun N-terminal kinase (JNK), and p38), NF-κB, and phosphoinositide 3-kinase (PI3K) [31]. The treatment of RAW264.7 cells with FX significantly reduced the phosphorylation of ERK and p38 in a concentration-dependent manner. However, JNK, NF-κB, and PI3K phosphorylation was not altered following FX treatment (Figure 6).

Previous studies demonstrated that nuclear factor erythroid 2-related factor 2 (Nrf2) is a negative regulator of osteoclastogenesis [32,33,34]. Nrf2 deficiency augments the RANKL-induced activation of ERK and p38 MAP kinases in mouse bone-marrow-derived osteoclast precursor cells [32]. Moreover, FX activates Nrf2 in nonbone and RAW264.7 cells [27,35,36,37]. We assessed the levels of phosphorylated Nrf2 and Nrf2 proteins in total cell extracts, and the nuclear and cytosolic fractions of cell lysates by Western blotting (Figure 7A). Nrf2 levels in total cell lysates were not affected by FX treatment. Expression of nuclear Nrf2 significantly increased, while cytosolic Nrf2 expression dose-dependently decreased. The proportions of phospho-Nrf2 expression in the nucleus compared to in the cytoplasm were significantly augmented by FX. This effect increased with concentration (Figure 7B). These results suggest that FX induces the dose-dependent phosphorylation and nuclear translocation of Nrf2 in RAW264.7 cells. 

## 3. Discussion

Several pharmacotherapeutic drugs such as bisphosphonate, estrogen, and anti-RANKL antibodies are used to treat osteoporosis. However, these drugs are commonly associated with side effects such as medication-related osteonecrosis of the jaw, atrial fibrillation, and esophageal cancer [38,39,40]. Moreover, osteoporosis is poorly treated globally despite therapeutic advancements [41]. The proportion of patients with a hip fracture who were prescribed bone-protective medication decreased from 40% to 21% between 2001 and 2011 [42]. This undertreatment and poor adherence to drugs may be due to fear of adverse effects [41]. Therefore, therapeutic agents with no or minimal side effects are required. The protective role of carotenoids on bone resorption has been recently gaining attention. Carotenoids present in fruits and vegetables, such as lycopene and β-cryptoxanthin, and marine carotenoid astaxanthin show inhibitory effects on osteoclastogenesis [19,20,43,44,45]. 

Prior studies investigated the therapeutic effect of FX—a marine carotenoid—on osteoclastogenesis. Kose et al. [21] investigated the therapeutic effect of FX on alveolar bone resorption in a ligature-induced periodontitis mouse model. FX treatment significantly reduced the number of RANKL-positive osteoclasts located in the resorption lacunae and increased serum bone-specific alkaline phosphatase levels. Das et al. [22] demonstrated that the treatment of osteoclasts differentiated from RAW264.7 cells with 2.5–5 μM FX inhibited RANKL-induced osteoclast differentiation by inducing apoptosis. However, FX did not exert cytotoxic effects in osteoblast-like cells at 2.5 μM. Although not the pure FX compound, extracts containing FX from *Sargassum fusiforme* suppressed osteoclast differentiation, promoted osteoblast formation [46], and exhibited antiresorptive effects in an ovariectomized mice model. These studies suggest that FX functions as a bone-protective agent in osteoclast-mediated skeletal disorders. Here, we showed that FX inhibits both the canonical RANKL-induced osteoclastogenesis and the RANKL-independent TNFα/IL-6-induced differentiation of osteoclasts (Figure 2). Moreover, FX significantly attenuated osteoclastic bone resorption pit in a dose-dependent manner (Figure 3). Unlike the results of Das et al., our results demonstrated that the inhibitory effect of FX at ≤5 μM is not mediated by the apoptosis of osteoclast precursor cells (Figure 1B).

To our knowledge, there is no study focusing on the signaling pathway underlying the bone-protective activity of FX. Osteoclast differentiation is a multistep process that involves cell proliferation, commitment, fusion, and activation [2]. During this process, RANKL interacts with RANK to recruit TNF receptor-associated factor (TRAF) adaptor protein and induce downstream targets such as NF-κB, MAPK (JNK, ERK, and p38), PI3K, and Akt [47,48]. Previous studies also reported that TNF-α and IL-6 can independently promote osteoclastogenesis in vitro of RANKL [7,8,49]. Moreover, TNF-α can activate various signaling pathways, including p38 MAPK, ERK, and NF-κB [48,50]. Our study confirmed that FX downregulates ERK and p38 in both RANKL-dependent and -independent pathways, but not JNK or PI3K (Figure 6). The pharmacological action of nitrogen-containing bisphosphonates is also mediated by the inhibition of the MEK/ERK pathway [51]. 

NFATc1 is characterized as a master molecule of RANKL-induced osteoclast differentiation and can autoamplify its own expression [52]. Previous studies demonstrated that NFATc1 binds to the promoter region of *MMP-9* and *DC-STAMP* in osteoclasts, and increases mRNA expression [53,54]. In line with these findings, our study demonstrated a significant decrease in *DC-STAMP* mRNA expression (Figure 4B) and production of MMP-9 (Figure 5) upon FX treatment, in addition to NFATc1 downregulation.

Nrf-2 is a transcription factor expressed in various cell types, and is known as a regulator of cytoprotective genes against oxidative and chemical injuries [55]. Under normal quiescent conditions, Nrf2 is tethered to cytoplasmic protein Keap1. However, in cells exposed to stressful stimuli, Nrf2 is released from Keap1 and activated via phosphorylation. Nrf2 phosphorylation is important for its stabilization, and phospho-Nrf2 preferentially translocates to the nucleus [55,56]. Nrf-2 can also regulate osteoclast formation and activity. The overexpression of Nrf2 suppressed RANKL-induced osteoclast differentiation by increasing the level of antioxidant enzymes and locally inducing nuclear Nrf2-attenuated osteoclastogenesis [33,34,55]. In the present study, FX treatment promoted the nuclear translocation and phosphorylation of Nrf2 in both RANKL-dependent and -independent pathways (Figure 7). Consistent with this finding, nuclear translocation and phosphorylation of Nrf2 by FX was reported in studies of human keratinocytes and ischemia/reperfusion-induced neuron cells [35,36]. As Park et al. reported, the induction of Nrf2 dramatically suppresses the transcriptional activity of NFATc1 [57], the activation of Nrf2 by FX in our findings suggests the decreased expression of NFATc1 by FX (Figure 4A). Conversely, previous studies using HepG2 cells under oxidative-stress conditions demonstrated that ERK or p38 kinase activation is required for drug-mediated Nrf2 translocation [58,59]. However, this study and a previous study with NRK-52E cells [60] showed that the treatment of cells with FX reduced phosphor-ERK/p38 levels and induced the nuclear translocation of phospho-Nrf2. The interaction between MAPK and Nrf2 pathways may vary depending on cell type, drug, or cell environment. 

Limitations exist in this study. Although murine RAW264.7 cells are frequently used as in vitro models of osteoclast differentiation, it is necessary to confirm the beneficial effect of FX in human osteoclast precursors. Furthermore, in vivo studies are required to examine the therapeutic potential of FX in disease models. Nonetheless, to our knowledge, this is the first study to clarify the molecular regulatory mechanisms of FX in osteoclast differentiation. 

We demonstrated that FX attenuates both RANKL-dependent and -independent osteoclastogenesis by downregulating ERK and p38 expression, and promoting the nuclear translocation of phospho-Nrf2 in RAW264.7 cells, as summarized in Figure 8. FX is confirmed to have no side effects and can be easily extracted from marine macro/microalgae. Therefore, FX represents a safe and inexpensive candidate drug for the treatment of various diseases accompanying the imbalance between osteoclasts and osteoblasts.

## 4. Materials and Methods

### 4.1. Cell Lines and Reagents

RAW264.7, a murine macrophage cell line, was purchased from the American Type Culture Collection (ATCC; Rockville, MD, USA). Human lung adenocarcinoma cell line NCI-H3122 was a kind gift from Professor Jong-Seok Lee (Seoul National University College of Medicine, Seoul, South Korea). Human peripheral blood mononuclear cells (PMBCs) were obtained from 4 anonymous donors (Koma Biotech, Seoul, Korea). Dulbecco’s Modified Eagle’s Medium (DMEM) and Minimum Essential Medium Eagle-Alpha Modification (α-MEM) were purchased from Welgene (Daegu, Korea). Fetal bovine serum (FBS) was obtained from Atlas Biologicals (Fort Collins, CO, USA) and penicillin–streptomycin from Gibco (Carlsbad, CA, USA). Recombinant mouse TNF-α, mouse IL-6, and human sRANKL were purchased from PeproTech (Rocky Hill, NJ, USA). CD14 MACS^®^ MicroBeads were purchased from Miltenyi Biotec Inc. (Auburn, CA, USA).

Antibodies against procaspase-3, caspase-3, PARP, cleaved-PARP, ERK, phospho-ERK, p38, phospho-p38, JNK, phospho-JNK, PCNA, PI3K, and phospho-PI3K were purchased from Cell Signaling Technology (Danvers, MA, USA). Anti-β-actin antibody was purchased from Enogene Biotech (New York, NY, USA), and anti-NFATc1, anti-phospho-p65, anti-Nrf2, and phospho-Nrf2 antibodies were from ABcam (Cambridge, UK).

ELISA kits for MMP-9 were obtained from R&D Systems (Minneapolis, MN, USA). The TRAP staining kit was acquired from Takara (Shiga, Japan), and the bone resorption assay kit from Cosmo Bio (Tokyo, Japan). The MTT assay kit was purchased from Sigma–Aldrich (St. Louis, MO, USA).

### 4.2. Cell Viability Test Using MTT Assay

Cell viability was determined using the MTT assay kit as per the manufacturer’s protocol. Cells were seeded into 48 well tissue culture plates at a density of 2 × 10^3^ per well in growth medium. After 24 h, logarithmic phase cells were incubated with different concentrations of FX for 5 days. Thereafter, MTT (5 mg/mL in PBS) was added to each cell. After 4 h, the medium was removed, and dimethylsulfoxide was added to solubilize MTT for an additional 4 h. After extraction with dimethylsulfoxide, optical density (OD) was measured at 495 nm. Percentage viability was calculated as (OD of drug-treated sample/OD of control) × 100.

### 4.3. Culture and Differentiation of Cell Lines

RAW264.7 cells were seeded in 48 well plates (2 × 10^3^ per well), and cultured in α-MEM containing 10% FBS and 1% penicillin/streptomycin at 37 °C in 5% CO_2_/95% O_2_ in a humidified cell incubator. The culture medium was replenished every 3 days. Osteoclast differentiation was induced either by stimulation with sRANKL (50 ng/mL) for 5 days or costimulation with TNF-α (50 ng/mL) and IL-6 (50 ng/mL). 

Cryopreserved human PMBCs were thawed and washed, and CD14-positive cells were isolated using anti-CD14 antibody-coated microbeads. CD14+ monocytes were seeded at 1.5 × 10^5^ cells/well in a α-MEM medium containing 10% FBS and M-CSF (50 ng/mL). After confirming that the cells remained attached the following day, they were treated with sRANKL (50 ng/mL) or TNF-α (50 ng/mL)/IL-6 (50 ng/mL) under conditions of 0, 1, 2.5, and 5 μM FX. The culture medium was replaced every 4 days, and multinucleated TRAP-positive cells were counted after 17 days.

### 4.4. Osteoclast Differentiation from RAW264.7 Cells and Osteoclast Activity Assays

To evaluate the direct effects of FX on osteoclast differentiation, RAW264.7 cells were treated with 0, 1, 2.5, and 5 μM FX under sRANKL or TNF/IL-6 stimulation. Cells were stained with TRAP after 4 days of stimulation, and TRAP-positive multinucleated cells were enumerated under a light microscope.

Osteoclast activity was determined by measuring the area of resorption pits using calcium phosphate-coated 48 well plates according to the manufacturer’s recommendation. RAW264.7 cells were washed once with α-MEM containing 10% FBS and seeded onto 48 well plates (2 × 10^3^ per well). The following day, cells were treated with FX under sRANKL or TNF/IL-6 stimulation. On Day 5, the pit area was measured after adding 5% sodium hydrochlorite along the plate wall to remove RAW264.7 cells, and after washing with water. After air drying, microscopic images of all fields were acquired, and the resorbed pit area per well was measured using ImageJ software (NIH, Bethesda, MD, USA).

### 4.5. Immunoblotting

To determine the expression of intracellular proteins, immunoblotting was performed with the aforementioned antibodies. For NFATc1 and Nrf2 expression, RAW264.7 cells were seeded onto a 6 well plate (2.0 × 10^4^ per well) in α-MEM containing 10% FBS. On the following day, 0, 1, 2.5, and 5 μM FX were added to the culture medium, and cells were grown under sRANKL or TNF-α/IL-6 stimulation for 4 days. For MAPK, p65, and PI3K expression, RAW264.7 cells were cultured in α-MEM containing 10% FBS with 0, 1, 2.5, and 5 μM FX for 4 days. Thereafter, cells were incubated with sRANKL or TNF-α/IL-6 in serum-free media for 30 min.

Total cell lysates were obtained using cold radioimmunoprecipitation assay (RIPA) buffer (25 mM Tris-HCl, pH 7.6; 150 mM NaCl; 1% NP-40; 1% sodium deoxycholate; 0.1% SDS). The crude extract was separated on 10% sodium dodecyl sulfate-polyacrylamide gel electrophoresis (SDS-PAGE) and transferred to polyvinylidene difluoride membranes. Protein bands were detected using an enhanced chemiluminescence system (Amersham Biosciences, Little Chalfont, UK). Nuclear extracts were prepared with the NE-PER Nuclear Cytoplasmic Extraction Reagent kit (Pierce, Rockford, IL, USA) according to the manufacturer’s instructions. The relative expression of each protein was determined by densitometric analysis using ImageJ software.

### 4.6. Reverse Transcription-Polymerase Chain Reaction (RT-PCR) and Real-Time PCR

The expression levels of osteoclast-related genes were measured using RT-PCR with specific primers. cDNA and target-specific primers were added to the power SYBR green PCR master mix (Applied Biosystems, Foster City, CA, USA). PCR cycling parameters were as follows: amplification (1 cycle at 50 °C for 2 min, 1 cycle at 95 °C for 10 min, and 40 cycles at 95 °C for 15 s and 60 °C for 1 min). Fold changes of gene expression were calculated with the ΔΔCt method using ribosomal protein S18 as the reference gene. Specific murine primers are summarized in Table 1.

### 4.7. ELISA

RAW264.7 cells were seeded onto a 6 well plate (2.0 × 10^4^ per well) in α-MEM containing 10% FBS. On the following day, 0, 1, 2.5, and 5 μM FX were added to the culture medium and incubated with sRANKL or TNF/IL-6 for 4 days. MMP-9 detection was performed using commercial ELISA kits according to the manufacturer’s instructions.

### 4.8. Statistical Analysis

All experiments were performed at least three times, and data are presented as mean ± SEM. Continuous variables were compared using Mann–Whitney U test. For dose-response analyses, the nonparametric Jonckheere–Terpstra trend test was performed. All data were analyzed using STATA^®^ SE, version 15.0 (StataCorp LLC, College Station, TX, USA). A *p* value < 0.05 was considered statistically significant.

## 5. Conclusions

FX inhibits osteoclast differentiation and bone-resorption activity through downregulating p38 and ERK, and promoting the nuclear translocation of phospho-Nrf2. The results of this study provide useful insight into the molecular mechanisms of FX action. Hence, FX could be used to treat bone diseases caused by excessive osteoclastic activity.

## Figures and Tables

**Figure 1 marinedrugs-19-00132-f001:**
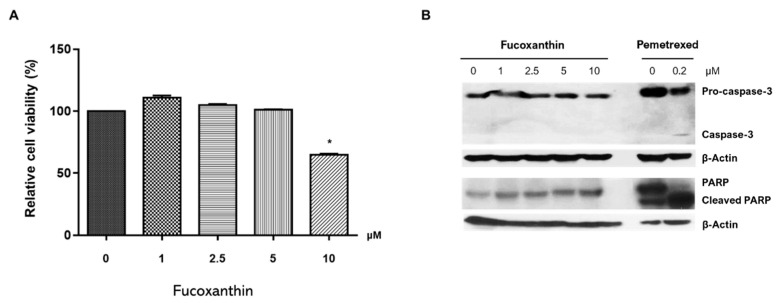
Effect of fucoxanthin (FX) on viability of RAW264.7 cells. (**A**) Cells treated with different concentrations of FX, and cell viability was determined using 3-(4,5-dimethylthiazol-2-yl) -2,5-diphenyltetrazolium bromide (MTT) assay. Data are representative of six independent experiments and are expressed as mean ± standard error of mean (SEM); * *p* < 0.05 versus FX-untreated cells (0 μM). (**B**) Procaspase-3 and poly ADP ribose polymerase (PARP) expression remained uncleaved upon treatment with ≤10 μM FX, unlike in pemetrexed-treated cells of lung-cancer cell line NCI-H3122.

**Figure 2 marinedrugs-19-00132-f002:**
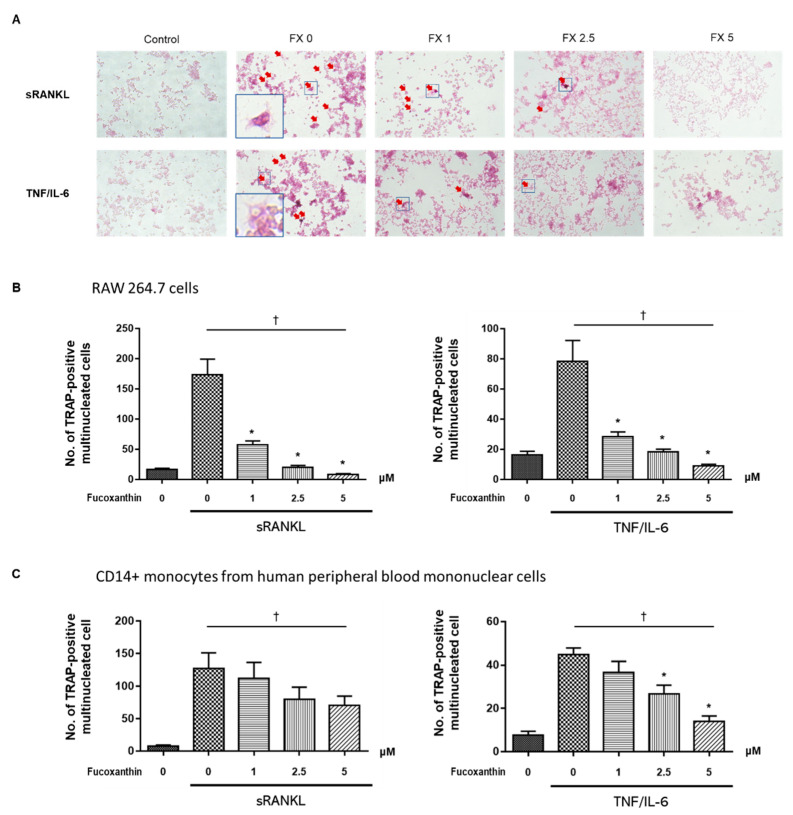
Soluble receptor activator of nuclear factor-κB (NF-κB) ligand (sRANKL)- or tumor necrosis factor (TNF)-α/interleukin (IL)-6-induced differentiation into osteoclast-like cells. (**A**) Representative microscopic images of tartrate-resistant acid phosphatase (TRAP) stained RAW264.7 cells (red arrows; original magnification, 100×). Blue box in bottom corner is a magnified photograph of the smaller boxed area (original magnification, 400×). (**B****,C**) Number of TRAP-positive multinucleated cells differentiated from (**B**) RAW264.7 cells and (**C**) human CD14+ monocytes decreased upon treatment with FX in a dose-dependent manner. Data are representative of three independent experiments and are expressed as mean ± SEM; * *p* < 0.05 versus FX-untreated osteoclast-differentiated cells; ^†^
*p* < 0.05 by Jonckheere–Terpstra test. FX, fucoxanthin.

**Figure 3 marinedrugs-19-00132-f003:**
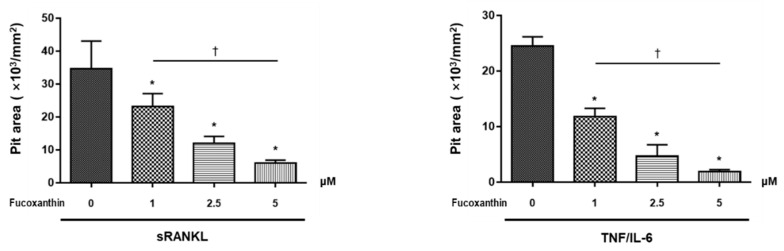
Effect of FX on osteoclast activity. Statistical differences of resorption pit area and trends tests are presented in histograms. Data are representative of three independent experiments and are expressed as mean ± SEM. * *p* < 0.05 versus FX untreated cells; ^†^
*p* < 0.05 by Jonckheere–Terpstra test.

**Figure 4 marinedrugs-19-00132-f004:**
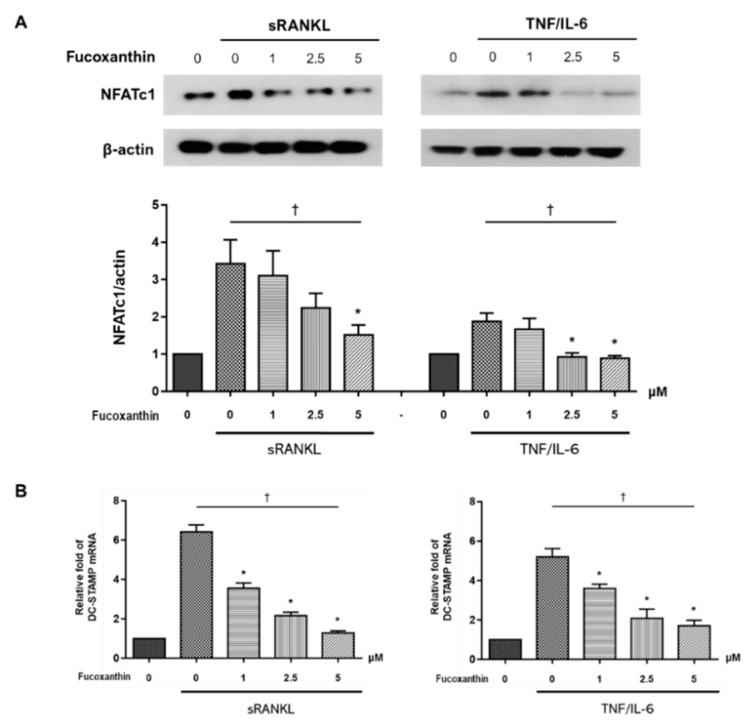
Effect of FX treatment on expression of osteoclast-specific markers. (**A**) Nuclear factor of activated T cell 1 (NFATc1) protein expression and (**B**) dendritic-cell-specific transmembrane protein (*DC-STAMP*) mRNA expression in sRANKL- and TNF/IL-6-stimulated RAW264.7 cells decreased upon treatment with FX in a dose-dependent manner. Data are representative of three independent experiments and expressed as mean ± SEM; * *p* < 0.05 versus FX untreated cells; ^†^
*p* < 0.05 by Jonckheere–Terpstra test.

**Figure 5 marinedrugs-19-00132-f005:**
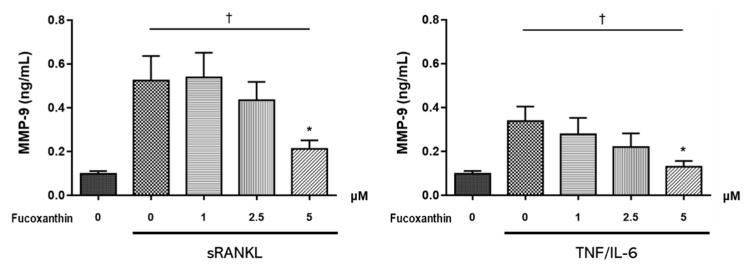
Effect of FX treatment on MMP-9 levels in culture supernatant of osteoclast differentiated RAW264.7 cells. MMP-9 concentration significantly decreased upon treatment with 5 μM FX. Data are representative of three independent experiments and expressed as mean ± SEM; * *p* < 0.05 versus FX untreated cells; ^†^
*p* < 0.05 by Jonckheere–Terpstra test.

**Figure 6 marinedrugs-19-00132-f006:**
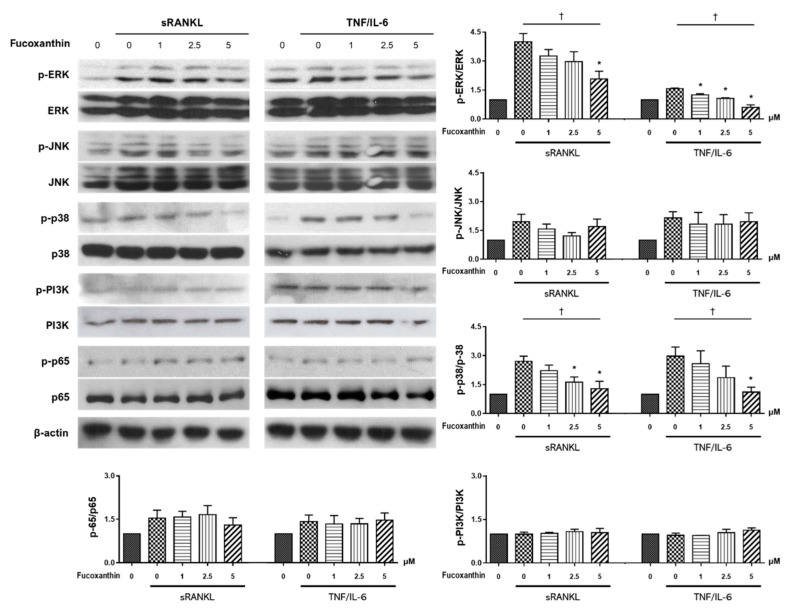
Effect of FX treatment on signaling pathways during osteoclastogenesis. FX inhibited extracellular signal-regulated kinase (ERK) and p38 activation in both RANKL- and TNF-α/IL-6- stimulated conditions. However, c-Jun N-terminal kinase (JNK), phosphoinositide 3-kinase (PI3K), and NF-κB levels were not significantly altered. Data are representative of five independent experiments and expressed as mean ± SEM; * *p* < 0.05 versus FX untreated cells; ^†^
*p* < 0.05 by Jonckheere–Terpstra test.

**Figure 7 marinedrugs-19-00132-f007:**
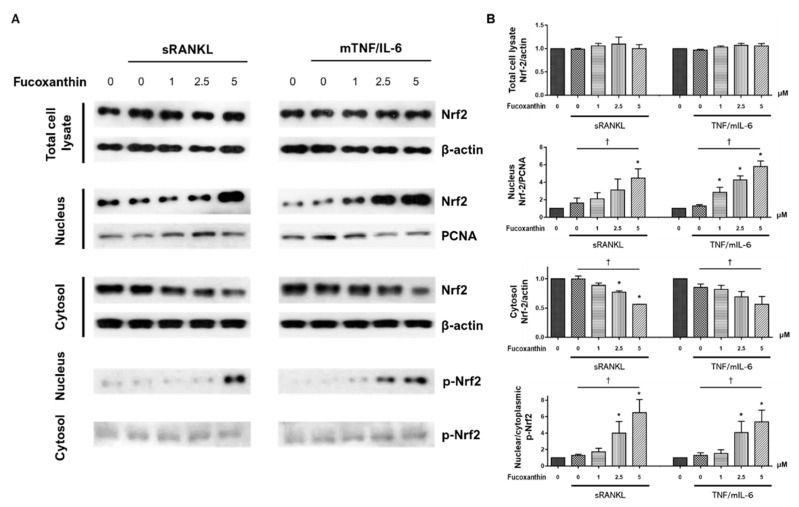
Effect of FX treatment on phosphorylated nuclear factor erythroid 2-related factor 2 (p-Nrf2) expression and nuclear localization of Nrf2 during osteoclast differentiation. Total cellular proteins were extracted from RAW264.7 cells and p-Nrf2 and Nrf2 expression were assessed by Western blotting. Cell lysates were fractionated into nuclear and cytosolic extracts, and identical experiments were performed. (**A**) Representative immunoblots and graphs for Nrf2 in total cell lysate, nucleus, and cytosol, and (**B**) nuclear/cytoplasmic p-Nrf2 from three independent experiments are shown. Data expressed as mean ± SEM; * *p* < 0.05 versus FX untreated cells; ^†^
*p* < 0.05 by Jonckheere–Terpstra test.

**Figure 8 marinedrugs-19-00132-f008:**
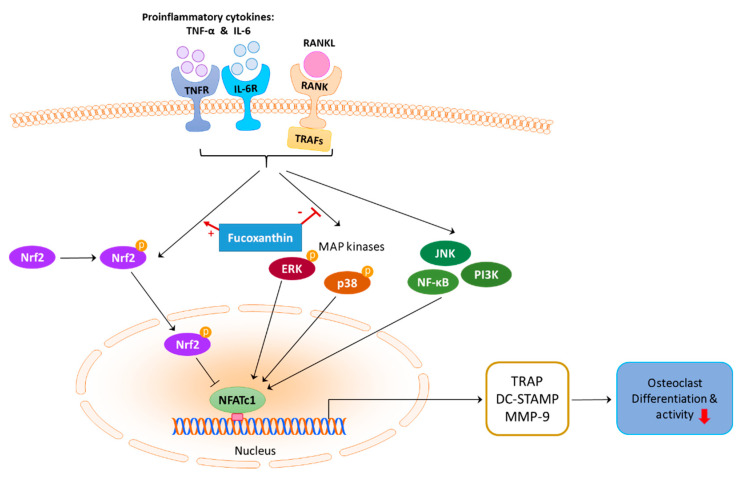
Signaling pathways and effects of FX during osteoclastogenesis. Inhibitory effect of FX is mediated by blocking the activation of ERK and p38, promoting Nrf2 nuclear translocation and phosphorylation, and subsequently downregulating NFATc1. Nrf2 induction was previously reported to suppress NFATc1 transcriptional activity [57].

**Table 1 marinedrugs-19-00132-t001:** Oligonucleotide primers used for RT-PCR.

Target Gene	GenBank Accession Number		Primer Sequence
*18S ribosomal RNA*	NR_003278	Forward	5′-GCAATTATTCCCCATGAA CG-3′
		Reverse	5′-GGCCTCACTAAACCATCCAA-3′
*DC-STAMP*	NM_029422	Forward	5′-TGCCAGGGCTGGAAGTTCAC-3′
		Reverse	5′-AAGGAGCTTCGCATGCAGGT-3′

## Data Availability

The data presented in this study are available on request from the corresponding author.

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
