# Peer review of "Fucoxanthin Suppresses Osteoclastogenesis via Modulation of MAP Kinase and Nrf2 Signaling"

_marinedrugs, 2021, doi:10.3390/md19030132_

Round 1
Reviewer 1 Report
The authors revealed that signaling pathways and effects of Fucoxanthin (FX) during osteoclastogenesis. The inhibitory effect of FX is mediated by blocking the activation of ERK and p38, promoting the nuclear translocation and phosphorylation of Nrf2, which has been previously reported to be an upstream regulator of RANKL-induced ERK/p38 cascade in osteoclasts, and subsequent down-regulation of NFATc1. This paper shows certain and significant data in osteoclastogenesis. However there are some defects in data and descriptions. Criticisms regarding this revised paper are discussed below.
Comments
The authors use murine macrophage cell line, RAW264.7 cells, as in vitro model of osteoclastogenesis. This model is useful for research for molecular mechanisms of differentiation from macrophasge to osteoclast. However the data using only one cell line should be confirmed by experiments using other models. There are in vitro murine and human osteoclastogenesis model using primary bone marrow derived progenitor cells such as CD34+/Lin- cells or macrophages. The additional data using primary osteoclast culture systems are necessary and valuable for confirmed evidences of inhibitory effect of FX against osteoclastgenesis.
Reviewer 2 Report
A natural carotenoid, Fucoxanthin (FX) has been known for its therapeutic potentials including bone disease, although its effects on osteoclasts remain unclear. In this study, Ha et al. examined the effect of FX on osteoclast-like cells using mouse cell line RAW264.7. The authors first determined cytotoxic dose of FX by MTT assay and found that 5 uM or lower dose of FX did not affect cell viability. They also described that 5 uM of FX did not affect cleavage of pro-caspase 3 or PARP. Next, the authors showed that FX inhibited RANKL- or TNFa/IL-6-induced osteoclast differentiation and resorption activities in a dose dependent manner. The authors examined the mechanisms underlying inhibitory effects of FX on osteoclast differentiation, and determined that FX inhibited induction of NFATc1, DC-STAMP and MMP-9 in a dose dependent manner. Furthermore, the authors showed that FX treatment inhibited RANKL- or TNFa/IL-6-induced phosphorylation of MAPK, p38 and ERK, both are required for proper osteoclastogenesis, while FX treatment augmented phosphorylation of Nrf2, a negative regulator of osteoclastogenesis in a dose dependent manner.
I would like to comment on this study.
- In biochemical experiments, the authors quantified and evaluated phosphorylation of MAPK and other essential signaling molecules required for osteoclastogenesis (Figure 6). I agree that quantification is important and graphs are helpful to see the results objectively. However, I do not see any change on phosphorylation of ERK and p38 by FX treatment on WB results (regarding p38, I do not see RANKL-induced phosphorylation at all). Meantime, I saw enhancement of phosphorylation of PI3K by FX, but corresponding graph did not show the enhancement. I do not think that the authors quantified WB band intensity accurately.
- The authors examined total and/or phosphorylated Nrf2 in Nucleus, Cytosol, and total cells (Figure 7), but I saw problems: in total cell extract, amount of total and/or phosphorylated Nrf2 did not change by FX treatment. However, it increased in Nucleus while nothing changes in cytosol by FX treatment. It should be decreasing in cytosol since in total cell extracts, amount of total and/or phosphorylated Nrf2 did not change by FX treatment. These results do not make sense, and I do not think that this experiment worked.
- It should describe more detail about Nrf2; What is it, what function is revealed in osteoclasts, what does phosphorylation mean, what does nuclear translocation mean, etc.
- The authors summarized results and drew a schematic diagram, but it seems confusing. What are the blue arrows with – symbol coming from near Nrf2-P? Does phospho-Nrf2 directly inhibit NFATc1? (according to ref. #32, Nrf2 deficiency does not change NFATc1 levels, though). Do TNF-a, IL-1, and IL-6 use the same cytokine receptor? Are these cytokines not involved in activation of MAPK and/or NF-kB? Is RANKL-RANK signaling not involved in phosphorylation of Nrf2?
- The authors examined pro-apoptotic effect of FX by examining caspase-3 and PARP, but these results are not clearly labeled: did the authors detect “Caspase-3” or “Pro-caspase-3”? Why are there two bands in PARP blotting?
- The authors showed that FX treatment dose dependently inhibited osteoclastogenesis. So, it is obvious that FX treatment inhibited pit formation (no osteoclasts, no pit formation). I suggest Figure 3 to move supplementary figures since this figure does not have significant meaning.
- Minor; in line 2, the word “Title” should be deleted.
Reviewer 3 Report
The manuscript is very interesting and obtained in it results clarify the molecular regulatory mechanisms of FX in osteoclast differentiation. FX has been confirmed to have no side effects, can be easily extracted from marine macro/microalgae and therefore, can be a safe, inexpensive candidate drug for the treatment of various diseases accompanying imbalance between osteoclasts and osteoblasts.
Introduction and Discussion paragraphs are strength parts of the paper. The first paragraph of the Discussion (together with Figure 8) may be moved to the last paragraph summarizing the study and may be also separated from indication of paper limitations, especially that almost the same sentences are written in the 5. Conclusions paragraph. It will be easily to follow the comparison of earlier published data with authors new results in the Discussion paragraph when other Figures will be mentioned in appropriate places for ex. Figure 2 in line 204, Figure 3 in line 205, Figure 4B in line 224, Figure 5 in line 225, Figures 6 and 7 in line 229. The authors indicated that in vivo results are required and it is also worth to mention the publication of Tomoyuki Koyama [Extracts of marine algae show inhibitory activity against osteoclast differentiation, Adv Food Nutr Res 2011;64:443-54] showing that the fucoxanthin-rich component from brown algae have suppressive effects against osteoclast differentiation. An extract of Sargassum fusiforme suppressed osteoclast differentiation and accelerated osteoblast formation in separate in vitro experiments. It also showed antiosteoporosis activity in in vivo experiments on ovariectomized mice by regulating the balance between bone resorption and bone formation.
Results paragraph is a weak part of the paper and need some corrections. Specific questions or comments are below:
- Figure 1 - Why results after 10 mM FX are not presented at Figure 1B? It will be better to show also the histograms of protein of interest per actin level like in Figure 4A especially if only this Figure 1B is mentioned in the Discussion paragraph.
- Figure 2 - What is seen on Figure 2A? or indicate the multinuclear cells by arrows. Why on Figure 2B after stimulation by sRANKL 2x more TRAP-positive cells are observed then after stimulation by TNF/IL-6? It will be more communicative if Figure 2A will be mentioned in line 91 and Figure 2B in line 93.
- Figure 3 - Why results at 0 mM FX without any stimulation are not presented? Why after stimulation by sRANKL the higher Pit area is visible then after stimulation by TNF/IL-6?
- Figure 4 - Why on Figure 4A after stimulation by sRANKL 2x higher protein expression is observed then after stimulation by TNF/IL-6?
- Figure 6 – Why after stimulation by sRANKL 2x higher level p-RRK/ERK is visible then after stimulation by TNF/IL-6? Why only p-PI3K is presented per actin level? Is p-p65/p65 is a NF-kB?
- It will be more communicative if Figure 7A will be mentioned in line 157 and Figure 7B in line 159.
- Figure 8 – Something is wrong with the arrow from NF-kB to NFATc1, Has nuclear p-NRF2 a direct influence on NFATc1?, Show JNK and PI3K on the scheme?
References no 5, 20, 21, 27, 32, 37, 43, 50, 56 have some page numbers omitted.
Round 2
Reviewer 1 Report
The authors revealed that signaling pathways and effects of Fucoxanthin (FX) during osteoclastogenesis. The inhibitory effect of FX is mediated by blocking the activation of ERK and p38, promoting the nuclear translocation and phosphorylation of Nrf2, which has been previously reported to be an upstream regulator of RANKL-induced ERK/p38 cascade in osteoclasts, and subsequent down-regulation of NFATc1. This paper shows certain and significant data in osteoclastogenesis. The additional data in this revised version are valuable for confirmed evidences of inhibitory effect of FX against osteoclastgenesis.
Reviewer 2 Report
The authors answered all my questions carefully. I'm satisfied with this paper.